# Exposure Assessment of Aflatoxin B1 through Consumption of Rice in the United Arab Emirates

**DOI:** 10.3390/ijerph192215000

**Published:** 2022-11-15

**Authors:** Nisreen Alwan, Haneen Bou Ghanem, Hani Dimassi, Layal Karam, Hussein F. Hassan

**Affiliations:** 1College of Health Sciences, Abu Dhabi University, Abu Dhabi 59911, United Arab Emirates; 2Nutrition Program, Natural Sciences Department, Lebanese American University, Beirut P.O. Box 13-5053, Lebanon; 3School of Pharmacy, Lebanese American University, Byblos P.O. Box 36, Lebanon; 4Human Nutrition Department, College of Health Sciences, QU Health, Qatar University, Doha P.O. Box 2713, Qatar

**Keywords:** rice, mycotoxins, ELISA, dietary intake, contamination

## Abstract

Rice is one of the most consumed staple foods worldwide and a major part of the diet for half of the global population. Being primarily cultivated in countries with warm and humid environments increases rice’s susceptibility for mycotoxins contamination, especially the hepatotoxic and carcinogenic aflatoxin B1 (AFB1). Since no study was published before on the exposure to AFB1 from consuming rice in the UAE, our study aims to assess the levels of AFB1 in rice marketed in the country and determine the estimated daily exposure of the population for this carcinogenic metabolite and its associated liver cancer risk. All white, brown, and parboiled rice brands available in the retail markets in the UAE were procured twice. Using an enzyme-linked immunosorbent assay (ELISA) method, AFB1 was detected in 48 out of 128 rice samples (38%). The average contamination ± standard deviation of AFB1 among positive samples (above the detection limit) was found to be 1.66 ± 0.89 μg/kg, ranging from 1 μg/kg (detection limit) to 4.69 μg/kg. The contamination level in all the samples was below the limit set by the Gulf Cooperation Council Standardization Organization (≤5 μg/kg), while 10 (20.8%) of the positive samples had a contamination level above the maximum limit set by the European Union (≥2 μg/kg). The moisture content in all the assessed samples was ≤14%. Furthermore, there was a significant difference in AFB1 between samples in both collections (*p*-value = 0.043). However, the rice type, grain size, packing country, packing season, country of origin, collection season, and packing to purchasing time had no significant effect on AFB1. The calculated mean daily exposure level of the Emirati population to AFB1 from consuming rice was 4.83 ng/kg.

## 1. Introduction

Mycotoxins are categorized as toxic secondary metabolites produced by various genera of fungi, such as *Aspergillus*, *Fusarium*, *Penicillium*, and *Alternaria* [1]. More than 400 types of mycotoxins have been reported, but the most studied ones worldwide are aflatoxins (AFs), Ochratoxin A, Zearalenone, Trichothecenes, Fusarium toxins, and Patulin, due to their significant impact on health and agriculture [2,3]. Aflatoxins are among the most toxic and carcinogenic secondary fungal metabolites produced by the Aspergillus species. In nature, there are 20 types of AFs identified; however, the major AFs are AFB1, AFB2, AFG1, and AFG2 [4].

Aflatoxin-producing fungi grow widely in diverse stages of the food chain, infecting various agricultural products such as peanuts, dried fruits, rice, cottonseeds, wheat, maize, and nuts, leading to significant agricultural losses [5]. The mean daily dietary exposure to AFs differs between developed and developing countries due to the growth-promoting factors of aflatoxin-producing fungi in the latter [6]. Among these fungal toxins, AFB1 is considered the most potent toxin and mutagen, classified by the International Agency for Research on Cancer (IARC) in 2002 as a Group 1 carcinogen [4,7]. Several diseases and health complications have been linked to the chronic exposure to AFB1, including hepatocellular carcinoma, male infertility, hepatic encephalopathy and Reye’s syndrome, pulmonary interstitial fibrosis, and impaired growth during sensitive developmental periods [5,8]. AFs are thermally resistant to various food processing methods. Their melting point exceeds 250 °C, and they remain stable at a pH range between 3 and 8, making it a complicated challenge to eliminate them from the food chain leading to an increased risk of contamination. Therefore, preventive safety and quality measures should be taken to avoid their existence in the first place [7].

Rice is grown in hundreds of countries worldwide. Asia is considered the main continent in rice consumption and production, accounting for more than 90% of the total global production [9,10]. According to FAO, in 2016, rice was grown by nearly two billion and consumed by over four billion people [10]. In the Middle East and North Africa (MENA), the population growth in the last few decades was coupled with increased consumption of staple foods, particularly rice [11]. Rice consumption in the United Arab Emirates has increased from 14,000 to 800,000 metric tons between 1970 and 2021. In 2020, the consumption of rice by the Emirati population decreased due to the pandemic. However, as a rebound effect, and since the tourism rates are improving, rice consumption in the UAE increased by 6.67% from 2020 to 2021 [12].

Rice is primarily cultivated in countries having warm and humid environments, which promote the growth of aflatoxin-producing fungi in rice, especially during cultivation and storage [7,13]. Many drying methods used by the farmers, such as sun drying, are insufficient to reduce the moisture content of rice before storage, which will further increase the risk of contamination by AFs. Environmental changes in the harvesting season, such as heavy rainfall and floods, will also worsen the situation [13]. The occurrence of AFs in rice, particularly AFB1, has been assessed and detected in multiple studies. The European Union (EU) in 2006 has set the maximum limits for the occurrence of AFB1 in rice as 2 μg/kg, while these limits were set as 5 μg/kg by the GULF standards for rice [9,14,15]. Different analytical methods such as liquid chromatography-tandem mass spectrometry (LC-MS/MS), thin-layer chromatography (TLC), enzyme-linked immunosorbent assay (ELISA), and high-performance liquid chromatography (HPLC), can be used to detect the presence of AFs in rice [9].

Due to the tropical desert climate in the UAE and the increased temperature and humidity levels throughout the year, the marketed rice in this country is at an increased risk of being contaminated by AFs, threatening the population’s health and safety. [16,17] measured the ochratoxin A and toxic metal levels in rice marketed in the UAE, respectively; however, no study has been published before on the exposure to AFB1 from consuming rice in this country. Therefore, our study aims to evaluate the safety of the marketed rice, in terms of AFB1, in the UAE and evaluate the daily exposure level to this toxin from consuming rice among its inhabitants.

## 2. Methods

### 2.1. Rice Samples Collection

Ninety rice brands from different types (white, brown, and parboiled) in the UAE market were identified. Out of them, 38 were collected twice, and the other 52 brands were available in the retail markets during the first or second collection, but not both, bringing the total number of collected bags to 128. Collection took place in March 2021 and June 2021. Rice bags were stored in the freezer (−18 °C) until analysis. Information on packages (type, country of packing, country of origin, date between packaging and purchasing, presence of a food safety management certification) was retrieved. The presence of a food safety management system was checked by contacting the rice facilities by email. Characteristics of the rice brands are presented in Table 1.

### 2.2. AFB1 Determination

Immunoassay-based techniques are based on the binding of antigen to antibody. In addition to that, they are cheap, simple, and sensitive. One example of immunoassays is the enzyme-linked immunosorbent assay (ELISA). This technique is used easily and has a high sensitivity, adaptability, high-throughput minimal sample extraction and sample volume need. In addition, the quantitative analysis is performed by an intermediate, which is a simple and rapid kit that that shows similar results to TLC and HPLC [18,19]. In our study, the RIDASCREEN AFB1 30/15 test kit (R-biopharm, Germany, Product No: R1211) was implemented. The kit has 96 wells. The kit protocol was followed. In brief, a ground sample was transferred into tubes with methanol. Tubes were vortexed and centrifuged. Then, a supernatant liquid was diluted with distilled water. Standard/prepared samples were added into wells, followed by enzyme conjugate and antibody. The plate was then shaken and incubated for half an hour at room temperature. Liquid was poured out of wells, and the holder was tapped upside down. The wells were loaded with a washing buffer, and then poured out. Substrate/chromogen was then added to the wells, and the plate was shaken and incubated at room temperature for 15 min. A stop solution was added to the wells, and the absorbance was evaluated at 450 nm using a spectrophotometer. An analysis was completed in 2 replicates. The AFB1 concentration was determined from the standard curve developed based on the absorbance of standards (0, 1, 5, 10, 20, and 50 μg/kg). The AFB1 concentration was read using the cubic spline function of the RIDA^®^SOFT Win (Art. No. Z9999) software. The dilution factor 10 resulting from the sample preparation has already been considered. Therefore, the aflatoxin B1 concentrations of samples can be read directly from the standard curve. The limit of detection (LD) was 1 μg/kg as per the kit manufacturer.

### 2.3. Moisture Determination

Two grams of ground rice were transferred to a crucible, which was then placed in an air-oven at 130 ± 3 °C for 60 min. The crucibles were cooled down in a desiccator and weighed. Moisture was determined from the weight difference of rice (AOAC 22.013).

### 2.4. Assessment of the Exposure to AFB1

The average consumption of rice in the UAE (g/day) was assessed using the total consumption of rice in the country in 2021, and the number of the population, in addition to the average weight of adults in that year. The total consumption of rice in the UAE in 2021 was 800,000 MT, while the size of the population in the UAE in 2021 was estimated to be 9.99 million people [20], and the average weight of adults in the UAE in 2021 was 76 kg [21].

Thus, rice consumption per capita in UAE = (Total consumption of rice in the UAE in 2021 (kg))/(number of population in 2021) = 800,000 MT × 1000 kg/9,990,000 = 80.8 kg/year/person, which is equivalent to 221 g/day/person.

Then, the exposure level to AFB1 from rice consumption was determined as follows [22]:(Contamination level (ng/g) × Amount consumed (g/day))/(Average Body Weight (kg))

### 2.5. Statistical Analysis

The concentration of AFB1 was obtained as a mean of 2 replicate measures. Using SPSS V27, the central tendency and measure of spread were assessed using mean and standard deviations. The mean difference between groups was tested using the independent *t*-test for the type of rice, grain size, packing season, country of packing, food safety management certification (FSMC), and common brands between both collections. However, the ANOVA F-test was used to study the difference in means among the time between packing and purchasing. All analyses were performed at a significance level, *p*, of <0.05.

## 3. Results

AFB1 was found in 48 out of 128 (38%) of the samples tested. The average contamination ± standard deviation of AFB1 among positive samples (above detection limit) was found to be 1.66 ± 0.89 μg/kg, ranging from 1 μg/kg (detection limit) to 4.69 μg/kg. The contamination level in all the samples was below the limit set by the Gulf Cooperation Council Standardization Organization (≤5 μg/kg), while 10 (20.8%) of the positive samples had a contamination level above the maximum limit set by the European Union (≥2 μg/kg). All the assessed rice samples from both collections had less than 14% moisture content.

Our study showed no statistically significant difference in the AFB1 levels between white/parboiled rice and brown rice (*p* = 0.202) and between long rice and short/medium rice grains (*p* = 0.415). Regarding the rice brands that were packed in the Fall/Winter seasons and those in the Spring/Summer seasons in the UAE, the levels of AFB1 were not statistically significant between them (*p* = 0.676). Additionally, there was no statistically significant difference in the contamination levels between rice brands packed in the UAE and those packed in other countries, including Thailand, India, Pakistan, China, Italy, and the USA (*p* = 0.42). Similarly, this was true for the rice brands from developing countries (India, Thailand, Pakistan, and China) versus developed ones (Italy and USA) (*p* = 0.105). Furthermore, there was no statistically significant difference in the AFB1 levels between the time of rice packing and purchasing (*p* = 0.418).

On the other hand, there was a borderline significance (*p* = 0.049) in the AFB1 levels between rice brands with FSMC and those without FSMC or with missing information about FSMC (*p* = 0.049). Furthermore, there was a statistical difference (*p* = 0.043) in AFB1 between collections 1 and 2 of the brands collected twice (Table 2).

## 4. Discussion

To our knowledge, this is the first study in the UAE that assesses the estimated daily exposure of the Emirati population to AFB1 from consuming rice and the associated risk of liver cancer from this carcinogenic metabolite. Only one study was conducted back in the 1990s that assessed the occurrence of AFB1 in rice; however, it did not evaluate the estimated daily exposure level or the related liver cancer risk of the population. Moreover, the rice samples in the previously conducted study were collected from households in Al Ain City and not from the retail markets. This study reported higher contamination levels of rice ranging from 1.2 to 16.5 μg/kg, exceeding the upper limits set by the EU and those of the GSO [23]. One possible explanation for the difference in the results between this study and ours is that people used to store large quantities of rice at that period, thinking that extended storage would enhance the rice flavor while increasing the risk of mold growth and AFB1 production. In addition, the samples were collected in the summers of 1992 and 1994, favoring the growth of molds in stored rice due to warm environmental conditions.

Compared to the studies conducted in the nearby countries, including Saudi Arabia, Turkey, Lebanon, and Tunisia, the contamination level of rice was the highest in the UAE. In Saudi Arabia, the contamination of rice by AFB1 ranged from 0.014 to 0.123 μg/kg [7]. Ref. [24] reported that the concentration of AFB1 from rice samples collected from five provinces in Eastern Turkey ranged from 0 to 1.86 μg/kg. Concerning Lebanon, a study was conducted in 2014 that assessed the population’s exposure to AFB1 from their total dietary intake. It showed that the concentration of AFB1 in rice and rice-based products ranged from 0 to 0.010 μg/kg, which is also below the maximum limit set by the EU and the limits set by LIBNOR [25]. Another recent study by [26] reported that AFB1 was found in all rice samples (100%). Mean concentration ± standard deviation was 0.5 ± 0.3 μg/kg, and contamination ranged between 0.06 and 2.08 μg/kg. Only 1% of the samples exceeded AFB1’s European Union (EU) limit (2 μg/kg). In Tunisia, AFB1 was not detected in any of the collected samples [27]. Our results were also higher than those reported in China, where the contamination level was 0.06–2.1 μg/kg, and of those in Canada, where 49.7% of samples were contaminated with a mean contamination level of 0.36 μg/kg [28,29]. However, our results were comparable with Malaysia and Germany. In Malaysia, the contamination levels of AFB1 in rice ranged from 0.68 to 3.79 μg/kg [30]. Additionally, in Germany, the prevalence of AFB1 in rice was found between 0 to 4.61 μg/kg [31].

On the contrary, the contamination levels of rice by AFB1 in the UAE were lower than those reported by several countries. For example, in Pakistan, 95.4% of the tested samples were contaminated, with a mean contamination level of 3.80 μg/kg exceeding the upper limit set by the EU [32]. Similarly, Ref. [33] reported that the mean contamination level of AFB1 in rice marketed in Vietnam is 3.31 μg/kg. In addition, our findings were lower compared to India and the Philippines. In India, the contamination levels ranged between 0 to 361 μg/kg, where 38.5% of the 1511 tested samples were contaminated with significant amounts of AFB1 [34]. With respect to the Philippines, the contamination levels were reported to be between 0 and 8.33 μg/kg [35]. The differences in the reported contamination levels in rice between different countries are due to several factors, including the type of rice, size of the grain, country of origin and packing (developing versus developed countries), seasonal variations, and the safety of the adopted harvesting and storage conditions.

The low moisture content (MC) in our rice samples, ranging from 7 to 14%, elucidated the low detected levels of AFB1 in our study, which fell below the maximum level of MC set for rice grains (≤14%) [14]. Studies have shown that storing rice for extended periods with an MC below 14% will decrease the risk of contamination by AFB1 by hindering the growth of aflatoxin-producing molds [36,37]. This shows that the collected rice samples from the markets in the UAE were dried adequately after harvesting and stored in safe environments, maintaining safe MC, and decreasing the risk of contamination compared to other countries. Our results are lower than the previously conducted study in the country, where the MC of the collected samples ranged between 5.7 and 15.3%, with long moldy grains being the most contaminated compared to the insect-damaged long and short grains with a mean MC of 14.8 ± 1.4% [23].

Although brown rice is at a higher risk of AFB1 contamination [35], in our study, there was no statistical significance between AFB1 levels in brown versus white/parboiled rice; however, the mean AFB1 in brown rice was higher than that in white rice. Ref. [35] reported that in the Philippines, brown rice was more contaminated with AFB1 compared to polished rice, where the contamination level in polished rice ranged from 0.025 to 2.7 μg/kg lower than that of brown rice (0.03–8.7 μg/kg). This was confirmed by another study conducted in Pakistan, where 52.5% of the tested brown samples were contaminated with AFB1 with a mean of 0.56 μg/kg. However, only 40% of the tested white rice was contaminated with a lower average contamination level of 0.49 μg/kg [38]. The difference between the contamination levels in different types of rice is attributed to the milling process used to produce white or polished rice, which was confirmed in several studies [35]. The production of regular-milled rice from brown rice significantly reduced the mean aflatoxin levels by 78%. In addition, the processing of rough rice to regular-milled then to well-milled rice significantly decreased AFs by 68 and 82%, respectively [35].

In addition to the type of rice, the grain size was also shown to impact AFB1 levels, where long grains are more susceptible to contamination due to their larger surface area. In our study, although the mean contamination level of AFB1 in long grains was higher than that in short/medium grains, the difference was not significant (*p* ≥ 0.05), which might be attributed to the strict measures in the UAE over rice imports. However, our results are consistent with the literature. The previously conducted study in the UAE showed that 64% of the long-grain rice samples were contaminated with AFB1 compared to only 32% of the short-grain samples. Additionally, out of the 250 samples of long grain rice, 155 samples had a contamination level above the maximum set by the EU (≥2 μg/kg), compared to only 79 out of the 250 short-grain rice samples [23]. Another study conducted in Vietnam showed that 24/71 long-grain rice samples were contaminated with AFB1 ranging from 0.45 to 9.40 μg/kg; however, the 5 tested short-grain rice samples were free from AFs [39].

There was no significant difference in the contamination level between rice packed locally versus in other countries. This can be attributed to the safe packing techniques followed by both the UAE and the countries from which the UAE imports rice. When assessing the impact of having an FSMC on the contamination level of rice by AFB1, our results confirm that adopting safe and proper agricultural and harvesting processes by following the HACCP, ISO22000, or FSSC22000 standards, as these systems include good manufacturing practices as part of their requirements, will reduce mold growth, decreasing the risk of AFB1 contamination [5,40]. However, in our study, the country of origin, whether developing, including India, Thailand, Pakistan, China, or developed, including USA and Italy, showed no significant difference between AFB1 levels. In his recent review about the worldwide occurrence of mycotoxins in rice, Ref. [41] claimed that the contamination of rice by mycotoxins is the highest in the developing countries that have the lowest economic resources. In addition to having high temperatures and humidity, these countries, including Southeast Asia and Sub-Saharan Africa, do not have the facilities to properly store food, leading to the increased production of AFB1 during storage. On the other hand, the developed countries have the ability to store food in controlled climate conditions and have access to federal regulatory bodies that implement safety standards and perform a continuous inspection on the domestic and imported foods. These factors lower the risk of contamination in developed countries [5]. Another reason why developing countries might be more susceptible to AFB1 contamination in their food, and feed in their environments, is the fact that these countries mostly have tropical and sub-tropical climates, which have been shown to be mostly at risk of AF contamination, since high temperature and humidity enhance the growth of mycotoxigenic fungi [7]. In addition to the above-mentioned routes of contamination, it was also shown that some insects transfer the aflatoxigenic fungi between different crops. These insects are also more widely available in warm and moist climates, further increasing the risk of contamination in such environmental conditions [39,42]. In Pakistan, it was reported that the high contamination level of rice by AFB1 (95.4% of the samples were contaminated with an average of 3.80 μg/kg) is due to the country’s tropical climate that favors the production of AFs [43].

Our study showed that the samples that were packed during the Spring/Summer seasons had a higher contamination level (2.05 ± 0.80 μg/kg) compared to those that were packed during the Fall/Winter seasons (1.99 ± 0.73 μg/kg). However, the results did not reach significance, reflecting proper storage conditions regardless of the season they were packed in (*p* = 0.676). The UAE has a desert climate associated with warm weather and high humidity, enhancing the growth of aflatoxigenic fungi during storage, especially in the dry and humid summers that last from April to September. On the other hand, among brands collected twice, the mean contamination level of AFB1 in the second collection (2.16 ± 0.52 μg/kg) was significantly higher than that in the first collection (1.86 ± 0.78 μg/kg) (*p* = 0.043). This can be attributed to the inconsistency in the good agricultural and manufacturing practices in the rice facilities.

To assess the impact of extended storage on AFB1 levels in rice in the UAE, the association of the period between packing and purchasing and the contamination level was studied, but our results were not significant (*p* = 0.418). The insignificant results might reflect the proper rice storage in the UAE, where the temperature and humidity of the stores are adequately controlled, decreasing the risk of contamination, independent of the storage period. Several studies confirmed that contamination levels increased with increased MC and storage periods [41]. Ref. [30] showed that the fungal distribution, and consequently the levels of AFB1 in rice samples in India, vary between different states and storage conditions. The rice samples placed in open storage exposed to rain showed the highest contamination levels reaching 308 μg/kg.

Our study was the first in the UAE that assessed the estimated daily exposure of the Emirati population to AFB1 from their staple food, namely, rice. Based on the reported AFB1 levels in our samples, our results have shown that the daily exposure to AFB1 from rice ranged from 0.52 to 13.6 ng/kg BW/day with an average exposure level of 2.93 ng/kg BW/day. Our results were lower than those reported in several countries. In Iran, the average exposure level to AFB1 was 3.49 ng/kg BW/day [44]. On the other hand, the exposure was much higher in Pakistan and Vietnam. Ref. [45] found that the daily exposure to AFB1 ranged from 19.1 to 26.6 ng/kg BW/day. Regarding Vietnam, the maximum exposure level was 296 ng/kg BW/day, posing a serious health threat for its population [33]. However, our results were higher than other countries. For example, in Sweden, the daily exposure ranged from 2 to 3 ng/kg BW/day, which is comparable to South Korea, 1.19–5.79 ng/kg BW/day, and the Philippines, 0.1–7.5 ng/kg BW/day [35,46,47]. Compared to Lebanon, the daily exposure of the Emirati population to AFB1 was higher than the Lebanese population, which was estimated to range between 0.63 and 0.66 ng/kg BW/day based on their total dietary intake [25]. Another recent study in Lebanon showed that the exposure to AFB1 from rice consumption was 0.1–2 ng/kg body weight/day [26]. However, it is noteworthy to mention that when comparing the daily exposure level to AFB1 between different countries, we should be careful because many variables influence the results. The different methodologies used in the studies to assess the contamination of rice and the daily intake of rice by the population play a key role in the obtained results. Additionally, the different consumption patterns of rice between the various countries influence the daily exposure to AFB1 because the daily exposure to this contaminant depends on the contamination level and the daily intake of rice. Moreover, the type of rice studied, and the country’s environmental conditions significantly affect the exposure level of the population.

Some physical, chemical, and biological treatments have been shown to eliminate aflatoxins partially or totally from contaminated foods and feed. The physical removal of the kernels that are damaged by molds has been shown to reduce AFs by 40–80% [48]. With respect to the chemical approaches such as sorbic acid, acetic acid, and hydrogen peroxide, it was shown that they have several drawbacks, including the formation of toxic residues. Some farmers tend to use antimicrobial drugs instead; however, the microbes have developed increased resistance to them, decreasing their effectiveness. Therefore, the farmers are having an increased tendency to use plant-based natural antifungal agents to avoid such problems [49]. Ref. [50] reported that the use of lemon and pomegranate peels showed significant antifungal properties against Aspergillus flavus. The addition of pomegranate peels to the rice stored at 25 °C with 18% moisture content, which is considered the optimal medium for mold growth, has shown a total inhibition of aflatoxins production for a four-month storage period. With respect to lemon peels, they showed similar effects but for a three-month storage period. This shows that renewable bio-sources can replace chemical treatments to safely store grains for extended periods.

To conclude, concerning the strengths of our study, this is the first of its kind in the UAE that assesses the estimated daily exposure to AFB1 from consuming rice. In addition, this is the first study to determine the levels of AFB1 in various rice brands in the country’s market using the ELISA technique, which is a highly sensitive and selective quantitative method. However, our study has some limitations that should be considered when interpreting our results. First, our small sample size for certain variables might have masked the true effect of these variables. Second, only 38 brands were found common between collections one and two, which might have also influenced the results’ significance when comparing the two collections. Additionally, the estimated daily intake level was assessed using available statistical data on the average Emirati population due to the absence of records, which might have under- or overestimated the exposure level to this carcinogenic metabolite and the associated liver cancer risk. Furthermore, due to the diversity in the UAE population, exposure to AFB1 by nationality must be determined, since rice consumption patterns differ among different groups. In addition, future studies must assess any AFB1 differences according to the food safety management certification (HACCP, ISO22000, FSSC22000 …) present in the facility [26]. Moreover, most of the collected rice samples were white grains, packed outside the country, and collected from reputable food stores, which might affect the generalizability of our results.

## 5. Conclusions

In conclusion, rice is one of the most consumed staple foods in the UAE. It is essential to maintain its safety to secure the health and nutrition of the Emirati population. This study detected AFB1 in 48 out of 128 rice samples (38%). The average contamination ± standard deviation of AFB1 among positive samples was 1.66 ± 0.89 μg/kg. While the contamination level in all the samples was below the limit set by the Gulf Cooperation Council Standardization Organization, 20.8% of the positive samples had a contamination level above the maximum limit set by the European Union. Our results indicated that exposure to AFB1 from rice in the UAE is a health concern, implying that several stakeholders in the country should be involved. To protect the Emirati population from this carcinogenic metabolite, an integrated collaboration between different sectors is required to take strict preventive measures by applying good agricultural and manufacturing practices at various stages of the food chain. In addition, the rice should be imported from reputable suppliers with continuous monitoring of its quality and safety by performing routine AF analyses. On a household level, consumers must store dry rice in well-sealed containers away from humidity and heat sources. Future studies should include unpacked rice to better assess the contamination of rice by AFB1 in the country. In addition, food frequency questionnaires must be collected from the population to better reflect their rice consumption patterns and health history.

## Figures and Tables

**Table 1 ijerph-19-15000-t001:** Characteristics of the rice samples.

Variable	N
**Rice Type ^a^**	
White/Parboiled	98
Brown	30
**Grain Size ^a^**	
Long	76
Short/Medium	52
**Packing Season (UAE as country of packing)**	
Fall/Winter	77
Spring/Summer	51
**Country of Packing ^a^**	
UAE	40
Other countries	79
**Country of Origin ^a^**	
Developing (India, Pakistan, Thailand, and China)	113
Developed (USA, Italy)	15
**Food Safety Management System ^a^**	
Presence	58
Absence/Information not available	70
**Time between packing and purchasing**	
1 to 9 weeks	9
10 to 19 weeks	48
20 to 29 weeks	29
30 weeks and above	42
**Collection**	
1	38
2	38

^a^ In nine rice bags, the country of packing was not indicated.

**Table 2 ijerph-19-15000-t002:** Effect of different independent variables on mean (μg/kg) AFB1 levels in rice samples.

Variable	Mean	SD	*p*-Value
**Rice Type ^a^**			
White/Parboiled	1.96	0.67	
Brown	2.21	0.99	0.202
**Grain Size ^a^**			
Long	2.2	0.69	
Short/Medium	1.97	0.45	0.415
**Packing Season (UAE as country of packing)**			
Fall/Winter	1.99	0.73	
Spring/Summer	2.05	0.8	0.676
**Country of Packing ^a^**			
UAE	1.95	0.81	
Other countries	2.08	0.77	0.42
**Country of Origin ^a^**			
Developing (India, Pakistan, Thailand, and China)	2.04	0.8	
Developed (USA, Italy)	1.8	0.32	0.105
**Food Safety Management System ^a^**			
Presence	1.99	0.84	
Absence/Information not available	2.53	0.69	0.049
**Time between Packing and Purchasing**			
1 to 9 weeks	2.17	0.97	
10 to 19 weeks	2.11	0.95	
20 to 29 weeks	2.03	0.75	
30 weeks and above	1.86	0.39	0.418
**Collection**			
1	1.86	0.78	
2	2.16	0.52	0.043

^a^ In nine rice bags, the country of packing was not indicated.

## Data Availability

Data sharing not applicable.

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
