# Peer review of "Exposure Assessment of Aflatoxin B1 through Consumption of Rice in the United Arab Emirates"

_ijerph, 2022, doi:10.3390/ijerph192215000_

Round 1

Reviewer 1 Report

The manuscript tackles the exposure of UAE inhabitants to AFB1 through the consumption of rice. This is a much needed topic to tackle in a hot and humid country, like UAE, where rice is a staple food commodity as well. The results are significant and of importance to policy makers and community at large.   My comments are the following:   1. References should be numerical. Please follow the journal instructions in this regard.   2. Please specify the month and year of the rice bags collection in section 2.1   3. Why did authors tackle AFB1 only? Why not OTA as well?   4. Why did not authors validate the results using HPLC, the golden method to analyze myvotoxins in cereals?   5. Why did authors exclude unpacked rice from the study?   6. In the "conclusion", please mention tips for the customers to avoid mold growth, and thus AFB1 secretion, in the rice at the household levels.   7. In the limitations paragraph, do authors consider using ELISA instead of HPLC as limitation?   8. UAE population is very diverse in terms of nationalities. Please add to the "recommendations for future studies" to investigate the consumption of rice patterns and associated exposure to AFB1 per nationality, gender, age group ... This is important as it is applicable to other neighboring Gulf countries   9. In table 1, under "country of packing", the total no of samples is 117, not 128, how come?!   10. In table 1, "time between packing and", there is a missing word here. Please correct.   11.How did authors identify the presence of a food safety management certification? Were the rice importers/facilities contacted?   12. In future studies, it would be interesting to see the AFB1 difference by food safety management system. FSSC22000 involves unannounced audits, unlike ISO22000. Is this difference making facilities with FSSC22000 certification more conforming regarding AFB1 contamination?  

Author Response

Thank you for the valuable comments, which were carefully addressed:

References should be numerical. Please follow the journal instructions in this regard.  

We will do it in the final draft since the references got edited in addressing the reviewer comments.

Please specify the month and year of the rice bags collection in section 2.1  

"Collection took place in March 2021 and June 2021" was added to 2.1

Why did authors tackle AFB1 only? Why not OTA as well?  

We already measured OTA in rice and published the work recently in IJERPH: https://pubmed.ncbi.nlm.nih.gov/36078789/

Why did not authors validate the results using HPLC, the golden method to analyze mycotoxins in cereals?  

ELISA technique is a highly sensitive and selective quantitative method. It was used in a plethora of studies published in top tier journals. This was mentioned  in the last paragraph of section 3.

Why did authors exclude unpacked rice from the study?  

In UAE, rice is supplied as packaged in general. Still, we stated in the recommendation for future studies to include unpacked rice

In the "conclusion", please mention tips for the customers to avoid mold growth, and thus AFB1 secretion, in the rice at the household levels.  

We added the following: "On household level, consumers must store dry rice in well-sealed containers away from humidity and heat sources."

In the limitations paragraph, do authors consider using ELISA instead of HPLC as limitation?  

We believe that using ELISA instead of HPLC is not a limitation as ELISA is a well-established technique. 

UAE population is very diverse in terms of nationalities. Please add to the "recommendations for future studies" to investigate the consumption of rice patterns and associated exposure to AFB1 per nationality, gender, age group ... This is important as it is applicable to other neighboring Gulf countries  

We added the following to the last paragraph of "conclusion" in the recommendations for future research part: " Furthermore, due to the diversity in the UAE population, exposure to AFB1 by nationality must be determined since rice consumption patterns differ among different groups"

In table 1, under "country of packing", the total no of samples is 117, not 128, how come?!  

Actually the total is 40+79=119. It is because that 9 rice bags were missing the country of packing

In table 1, "time between packing and", there is a missing word here. Please correct.  

"and purchasing". Table was fixed

How did authors identify the presence of a food safety management certification? Were the rice importers/facilities contacted?  

We checked the info on their package or their websites. if info was missing, we reached the rice facilities by email. This was added to section 2.1

In future studies, it would be interesting to see the AFB1 difference by food safety management system. FSSC22000 involves unannounced audits, unlike ISO22000. Is this difference making facilities with FSSC22000 certification more conforming regarding AFB1 contamination?  

We added this to the recommendations for future research at the end of the "discussion."

Reviewer 2 Report

Alwan et al. have conducted a study aimed at assessing the level of AFB1 in rice marketed in the United Arab Emirates and determining the estimated daily exposure of the population to this carcinogenic metabolite and the associated risk of liver cancer.

  1. Please explain the principle analysis regarding AFB1 determination using the Enzyme-Linked Immunoassay (ELISA) technique.
  2. How accurate is the determination result using this method compared to other methods? The results of determining AFB1 from the literature used different methods.
  3. Please explain the result data of moisture determination with AFB1 determination.
  4. Conclusions must answer the research objectives in detail. The conclusion of this manuscript needs to be revised again.
  5. What is the incidence rate in the United Arab Emirates country's patients concerning the number of detected liver cancer patients? Is the number increasing every year? Can it be related to the results of determining AFB1 in the rice consumed? What is the data? Who is most vulnerable to this disease, children, adults, men, or women?
  6. Please note that the writing of the literature is not consistent.

Author Response

Thank you for the valuable comments, which were addressed carefully.

Please explain the principle analysis regarding AFB1 determination using the Enzyme-Linked Immunoassay (ELISA) technique. How accurate is the determination result using this method compared to other methods? The results of determining AFB1 from the literature used different methods.

The following was added to section 2.2:

Immunoassay-based techniques are based on the binding of antigen to antibody. In addition to that, they are cheap, simple, and sensitive. One example of immunoassays is enzyme-linked immunosorbent assay (ELISA). This technique is used easily and have high sensitivity, adaptability, high-throughput minimal sample extraction and sample volume need. Also, the quantitative analysis is performed by an intermediate, which is a simple and rapid kit that that shows similar results to TLC and HPLC. (Iqbal et al., 2014; Pereira, Fernandes, & Cunha, 2014).

Please explain the result data of moisture determination with AFB1 determination.

As indicated in the "Results and Discussion" section, "all the assessed rice samples from both collections had less than 14% moisture content"; thus, we could not use moisture as independent variable in our statistical analysis.

Conclusions must answer the research objectives in detail. The conclusion of this manuscript needs to be revised again.

Conclusion was revised

What is the incidence rate in the United Arab Emirates country's patients concerning the number of detected liver cancer patients? Is the number increasing every year? Can it be related to the results of determining AFB1 in the rice consumed? What is the data? Who is most vulnerable to this disease, children, adults, men, or women?

The following paragraph was added to the "Introduction":

In UAE, hepatocellular carcinoma is increasing gradually from 1.1 cases per 100,000 population in 1990 to 2.37 cases per 100,000 in 2019. This rise is in parallel with increasing prevalence of obesity, diabetes and unhealthy dietary habits. Majority of cases (74%) are due to hepatitis B and C viral infections, while other causes of liver cancer have a relatively minor contribution. Highest incidence was reported at the age of around 80 to 90 years, while the male to female ratio is 2.7:1 (Jawad Hashim et al., 2021).

Reviewer 3 Report

Dear authors,

Comments and revision recommendations are attached.

Please revise the manuscript accordingly.

Reviewer.

Author Response

Thank you for the valuable comments which were carefully addressed.

Abstract: remove this informational paragraph

The underlined paragraph represents the bulk of our results. It presents the effect of different independent variables on AFB1 in our rice samples. We believe that this information must be presented in the abstract to highlight the study findings. If the editor supports the comment of the reviewer, we will delete this paragraph.

MT: spell out as it is used for the first time

"MT" was replaced with "metric tons"

2021(Hamza, 2020): spacing request

Done

The European Union (EU) in 2006 has set the maximum limits for the occurrence of AFB1 in rice. The maximum level of AFB1 in rice ready for human consumption is 2 μg/kg. However, the limits in rice before human consumption are different, with 5 μg/kg (Ali, 2019; Castaño et al., 2017). According to the GULF standards for rice, the levels of AFB1 should fall below or equal to 5 μg/kg (GSO, 2013). 

Rephrase these sentences to avoid information redundancy

These sentences were rephrased to:

"The European Union (EU) in 2006 has set the maximum limits for the occurrence of AFB1 in rice as 2 μg/kg, while these limits were set as 5 μg/kg by the GULF standards for rice (GSO, 2013; Ali, 2019; Castaño et al., 2017). "

"Hassan et al. (2022) measured the ochratoxin A levels in rice marketed in UAE; however, no study was published before on the exposure to AFB1 from consuming rice in this country" / mention this information gap in the abstract

Done

"AFB1 wise" / consider revising

Changed to "in terms of AFB1"

-18oC / space out

Done

"thai" / consider revising "developed based"

"that" changed to "developed based"

50μg/kg / space out

Done

60min / space out

Done

" The limit of detection (LD) was 1 μg/kg as per the kit manufacturer" / Consider mentioning the dilution protocol

"The dilution factor 10 resulting from the sample preparation has already been
considered. Therefore, the aflatoxin B1 concentrations of samples can be read
directly from the standard curve" was added at the end of section 2.1

Exposure to AFB1 assessment / consider revising the title

Changed to "Assessment of the exposure to AFB1"

numberof population / space out

Done

Consider re-organizing the information by giving each result followed by discussion

Done. We split "Results and Discussion" into 2 sections for better presentation of the study findings

Consider providing moisture data in bar charts. Consider tabulating the moisture data.

We tried, but since moisture data are very close among all samples, and all moisture values were conforming, we could not use moisture as independent variable in our statistical analysis

"128 (38%) of the samples", consider revising ... "tested"

Changed to "128 (38%) of the samples tested"

"no statistical difference" / consider revising (3 times in the paragraph)

Changed to "no statistically significant difference"

"FSMC" spell out the full name

"food safety management certification" was added to the first time we used "FSMC" in 2.5 Statistical Analysis 

"about it" / unclear what is "it"

"It" got changed to "FSMC"

Table 1: consider using mean AFB1 levels

Done

Table 1: specify the mass unit

"μg/kg" was added

Table 1: country of packing / numbers do not add up to 128

Correct. Numbers add up to 119. As indicated in the footnote, 9 rice bags were missing the country of packing

Table 1: collection / numbers do not add up to 128

Correct. As indicated in 2.1, only 38 brands were found in both collection dates

"Only one study was conducted back in the 1990s that assessed the occurrence of AFB1 in rice; however it did not evaluate the estimated daily exposure level or the related liver cancer risk of the population" / Check lines 92-94 to ensure information consistency

Indeed, one study was carried out previously in UAE on levels of AFB1 in rice in UAE, but it did not calculate the exposure level to AFB1 from consuming rice. 

GCC Standardization Organization (GSO) / consider mentioning this abbreviation in L85 and use GSO thereinafter 

Done

"hot" / consider using "warm"

Done

"2μg/kg" / space out request

Done

"sound" / check for wording correctness and accuracy

"sound" was deleted

"Although brown rice is at a higher risk of AFB1 contamination" / please provide a citation

Done - (Sales and Yoshizawa, 2005)

"confirmed in several studies" / provide citations

Done - (Sales and Yoshizawa, 2005)

"When assessing the impact of having a FSMC on the contamination level of rice by AFB1, our results showed borderline significance. The mean contamination level in the brands that have a FSMC is 1.99±0.84 μg/kg, which was lower than those that do not have a FSMC or have missing information about it, 2.53±0.69  μg/kg (p=0.049).

Consider removing this repeated information

Deleted

"Our results confirm that adopting safe and proper agricultural and harvesting  / processes by following the HACCP, ISO22000, or FSSC22000 standards

Explain how results relate to standards

Text edited to - our results confirm that adopting safe and proper agricultural and harvesting processes by following the HACCP, ISO22000, or FSSC22000 standards, as these systems include good manufacturing practices as part of their requirements

including, including / Remove including

Done

between the results / specify what results are we referring to

"results" changed to "AFB1 levels"

Consider including Table 1 here

Actually, we inserted Table 1 when it was first mentioned in the text. If the editor endorses the reviewer comment, we will change it to the location that the reviewer proposes

Our study is the first in the UAE that assess the estimated daily exposure of the Emirati population to AFB1 from their staple food, rice / Repeated information

This sentence was deleted

Due to the country's tropical desert climate, the rice market in the UAE is at increased risk of AFs contamination / do your results support these statements as UAE seasons do not have significant effect (Table 1)

This sentence was deleted

References / need to keep format consistency

We will have a through review of the "References" list formatting if our manuscript is accepted for publication

Round 2

Reviewer 2 Report

Since the author corrected and added the data I suggested, I recommend publishing the article.

Author Response

Thank you much for your positive response.

Reviewer 3 Report

Dear authors,

Attached contains revision recommendations.

Reviewer.

Author Response

Thank you for the valuable comments, which were all carefully addressed.

L29-30: You don’t have to mention this information here; move this to the discussion.

Thank you for the comment. The following statement was deleted from the abstract: "Presence of a food safety management system in place resulted in a borderline significance between certified brands (1.99±0.84 μg/kg) and not certified ones (2.53±0.69 μg/kg) (p-value=0.049)" (L29-30).

L34. Mention that this is daily exposure…

Thank you for the comment; "daily" was added to "mean exposure" (L34)

L62-67: Remove this paragraph as it has no direct connection with AFB1

Thank you for the comment. We added this paragraph upon the request of another reviewer. It got deleted in light of your comment (L62-67)

L109-110:  Remove the dots; list out all packaging information. 

Thank you for the comment. We removed the dots and listed all packaging information, i.e. "type, country of packing, country of origin, date between packaging and purchasing, presence of a food safety management certification" (L109-110)

L135 - Revision request. 

Thank you for the comment. "irectly" was fixed to "directly" (L135)

L172. p<0.05; the p has to be italicized; revise the p throughout the manuscript

Thank you for the comment. All "p" was italicized throughout the manuscript.

This table is full of shrunk data. The data could be displayed in separate
tables or figures to improve each data's comprehensiveness and the reader's comprehension and enable the reviewer's validation of data

Thank you for the comment. We split Table 1 into 2 tables. We added as well the raw results as supplementary materials.

L374 - Do your findings support this statement? Did you perform a quantitative study figuring out AFB1 amount and cancer development? Or, could you provide literature information that describes this connection? 

Thank you for the comment. We deleted the statement: "and the associated liver cancer risk" (L374)

Please provide a reference(s) to this statement - L388

Thank you for the comment. "Hassan et al., 2022" was added (L388)

...reflect their rice consumption patterns and health history (L409)

Thank you for the comment. "and health history" was added (L409)